# Evaluation of Nuclear Cataract with Smartphone-Attachable Slit-Lamp Device

**DOI:** 10.3390/diagnostics10080576

**Published:** 2020-08-09

**Authors:** Hiroyuki Yazu, Eisuke Shimizu, Sho Okuyama, Takuya Katahira, Naohiko Aketa, Ryota Yokoiwa, Yasunori Sato, Yoko Ogawa, Hiroshi Fujishima

**Affiliations:** 1Department of Ophthalmology, Tsurumi University School of Dental Medicine, Kanagawa 230-0063, Japan; go_my_way1987@yahoo.co.jp (S.O.); katahira-t@tsurumi-u.ac.jp (T.K.); fujishima117@gmail.com (H.F.); 2Department of Ophthalmology, Keio University School of Medicine, Tokyo 160-0016, Japan; ophthalmolog1st.acek39@gmail.com (E.S.); nao.nao.pao.pao@gmail.com (N.A.); yoko.z7@keio.jp (Y.O.); 3OUI Inc., Tokyo 160-0022, Japan; ryota.yokoiwa@gmail.com; 4Department of Preventive Medicine and Public Health, Biostatistics at Clinical and Translational Research Center, Keio University School of Medicine, Tokyo 160-0016, Japan; yasunori.sato@keio.jp

**Keywords:** cataract, portable, recordable, conventional slit-lamp microscope, smart eye camera

## Abstract

*Background:* Visual impairments and age-related eye diseases need to be detected and treated in a timely manner. However, this is often hampered by lack of appropriate medical equipment. We have invented a portable, recordable, and smartphone-attachable slit-lamp device, called the Smart Eye Camera (SEC). The aim of this study was to compare evaluating nuclear cataract (NUC) between the SEC and the conventional, non-portable slit-lamp microscope. *Methods:* A total of 128 eyes of 64 Japanese patients (mean age: 73.95 ± 9.28 years; range: 51‒92 years; female: 34) were enrolled. The NUC was classified into four grades (grade 0 to 3) based on three standard photographs of nuclear opacities according to the WHO classification by ophthalmologists. An ophthalmic healthcare assistant (non-ophthalmologist) filmed the eyes in video mode by the SEC and an ophthalmologist graded the NUC. Grade correlation and inter-rater reproducibility were determined. *Results:* NUC grading by the two approaches correlated significantly (both eyes: r = 0.871 [95%CI: 0.821 to 0.907; *p* < 0.001]). Inter-rater agreement was high (weighted κ = 0.807 [95%CI: 0.798 to 0.816; *p* < 0.001]). *Conclusions:* This study suggests that the SEC is as reliable as the conventional non-portable slit-lamp microscope for evaluating NUC.

## 1. Introduction

A slit-lamp microscope is an essential medical instrument in the field of ophthalmology [1]. This device irradiates the anterior segment of the eyes with a slit-light, and the reflection thereof is observed with its biomicroscope function [2]. However, most conventional slit-lamp microscopes are not portable, and patients have to visit the ophthalmologist for these examinations. Therefore, some patients (e.g., bedridden patients, children, the elderly, and infected patients) have less opportunity to undergo ophthalmology examinations. Although there are some portable slit-lamp microscopes, these devices are not recordable. An external camera is needed to record the anterior eye images, but it requires one to alter the instrument larger and heavier. Moreover, due to the large size and heavy weight of the conventional microscope to be used with one hand, it is difficult to use the microscope outside of the eye clinic which complicates validation between conventional non-portable and portable slit-lamp microscopes.

Blindness is increasing globally. It is estimated that 36 million people are now blind, and this number will increase to 115 million in the next 30 years [3]. The main cause of global blindness is cataracts. They account for 55% of blindness in adults older than 50 years globally [4]. Cataracts are the most common age-related ophthalmological disease, and they are diagnosed using a reflection of the thin slit-light from the slit-lamp microscope [5]. Blindness caused by cataracts is preventable by surgery [6]; nevertheless, lack of medical resources especially in undeveloped countries often hamper appropriate intervention [4,6,7]. An appropriate medical invention to detect cataracts could be the solution for this problem.

To address this issue, we have previously invented a portable and recordable slit-lamp device, the “Smart Eye Camera” (SEC), which convert the smartphone light source to the light needed for the ophthalmology diagnosis. In addition, SEC has a recording function using smartphone camera and was proven to be safe and feasible in an animal model [8]. We added slit-light converter to make a slit-beam thin enough to diagnose the cataracts in direct focal illumination method [9]. Here, we evaluated and compared the effectiveness and diagnostic performance for cataract diagnosis between SEC and conventional slit lamp microscope.

## 2. Materials and Methods 

### 2.1. Study Design

In this prospective study, 128 eyes of 64 individuals (30 males and 34 females) were enrolled. 4 eyes were excluded because the complete data was missing. The mean age of participants was 73.95 ± 9.28 years. In the current study, 110 of 124 eyes were phakic, 14 of 124 eyes were pseudophakic, and no aphakic eye was observed (Figure 1). We screened all patients who visited the Department of Ophthalmology, Tsurumi University School of Dental Medicine from July to September 2019. The patients who matched our inclusion criteria were recruited for the study. The inclusion criteria were (1) Japanese adult males and females (older than 20 years) who had been diagnosed as nuclear cataract (NUC), (2) cases with good mydriasis (>6 mm) to allow for better evaluation of the crystalline lens, and (3) no obvious ocular surface infections and/or inflammations. Patients who had at least one of the following were excluded: (1) Both pseudophakic or aphakic eyes, (2) Other types of cataract (i.e., cortical cataract [COR], anterior subcapsular cataract [ASC], and posterior subcapsular cataract [PSC]), (3) Severe corneal disease that adversely affected cataract grading (e.g., bullous keratopathy, band keratopathy, and corneal opacity), (4) Lack of data or patients who did not provide written informed consent. In total, 124 eyes in 62 cases were enrolled. The NUC grading of these eyes was evaluated with the portable and recordable Smart Eye Camera (SEC; OUI Inc., Tokyo, Japan) and with a conventional slit-lamp microscope (SL130, Carl Zeiss AG, Oberkochen, Germany).

This prospective study adhered to the tenets of the Declaration of Helsinki. All procedures were performed in compliance with the protocol approved by the Institutional Ethics Review Board of Tsurumi University Dental Hospital (approval number, 1634, 15 March 2019). Patient data were anonymized before access or analysis.

### 2.2. Conventional Non-Portable Slit-Lamp Microscope and SEC Examination

We used the SL130 as the conventional, non-portable slit-lamp microscope. This instrument is widely used for screening and diagnosis of several ocular diseases. For a comparison, the portable SEC slit-lamp device was selected. SEC is a smartphone attachment that fits above the light source and camera lens of a smartphone (Figure 2). The device has been approved as a medical device in Japan (Japan Medical Device registration number: 13B2X10198030101), and this is able to convert the light source of the smartphone to a thin slit-light, allowing the light to reach the crystalline lens. The slit-light of the SEC is unmovable as it uses the light source of the smartphone itself. Moreover, the slit-light angle is fixed by 40 degree which commonly used by the direct focal illumination method. To convert the light source of the smartphone to the 0.2 to 1.0 mm-width slit-light, a cylinder lens made from acrylic resin was attached in front of the light source. Above the camera lens of the smartphone, there is a removable convex macro lens to adjust the focus to the anterior segment of the eye. The frame was produced using a 3D printer (Multi Jet Fusion 3D Model 4210; Hewlett-Packard Company, Palo Alto, CA, USA) using polyamide 12. For the current study, an iPhone 7 (Apple Inc., Cupertino, CA, USA) was used as the camera and the light source. The resolution of the video was set at 1080 p and 30 frames per second, which equates to 2.1 megapixels (2,073,600 pixels) per file.

### 2.3. Cataract Evaluation by the Conventional Slit-Lamp Microscope and the SEC

After dilating the pupil, the recruited cases were examined using both conventional slit-lamp microscope and SEC. To minimize selection bias in terms of the order of the instrument used, we randomized all cases using a table of random digits. First, three ophthalmologists (H.F., H.Y., and S.O.) used the conventional slit-lamp microscope to examine the grade of the cataract. Second, a single orthoptist (T.K.) used the SEC to record a video of the cataract according to an instruction by an ophthalmologist (H.Y.). These examinations were all performed under a darkroom as well as an ordinary *light* environment in the ophthalmology department. Third, the images were documented in the medical records after filming. Finally, on another day, cataract grading was performed by a single ophthalmologist (H.F.) blinded to the patients’ information. The grading of the cataract by both the conventional slit-lamp microscope examination and the SEC examination was based on the NUC grading system by the WHO Cataract Grading Group [10]. When the diagnosis was different among the ophthalmologists, final diagnose of the NUC grading were decided by the majority consensus.

### 2.4. Data Analysis

Routine examinations, including a refraction test, visual acuity, intraocular pressure measurement using a non-contact tonometer, and fundus examination, were conducted for all cases, in addition to the clinical ocular evaluation using the SEC and conventional slit-lamp microscope. Visual acuity was measured using a standard Snellen chart, and the BCVA with spectacle correction was recorded. The results were measured in decimal acuity and converted to the logarithm of the minimal angle of resolution (logMAR) units using a visual acuity conversion chart (Appendix A).

### 2.5. Statistical Analysis

All data were analyzed using SPSS software (IBM SPSS statistics ver. 25; IBM Corp, New York, NY, USA) and Prism software (ver. 6.04 for Mac; GraphPad Software Inc, San Diego, CA, USA). It was not appropriate to pre-define a sample size, as this is a comparison study between a conventional instrument and a new device. Therefore, we selected all of the patients who matched our inclusion criteria within a certain period of time. Mann‒Whitney’s U test was performed to compare the NUC grade differences between conventional slit-lamp microscope and the SEC. To assess the reproducibility of the cataract grading by the two devices (conventional slit-lamp microscope and SEC), weighted kappa statistics were selected. Moreover, Spearman’s correlation coefficient was used to assess the correlation of the cataract grading evaluated using the 2 devices. Data were presented as adjusted means ± 95% confidence intervals (CI), ± standard deviation (SD), or ranges. P-value < 0.05 was considered to indicate statistical significance.

## 3. Results

### 3.1. Demographics of the Subjects

The mean severity of the NUC grading of the cataract was 1.84 ± 0.82 based on the evaluation by conventional slit-lamp microscope and 1.92 ± 0.82 by the SEC (*p* = 0.47, Table 1). There were no cases with NUC grade 0, which was not significant NUC formation, and grade 9, which was very advanced cataract that failed to provide adequate posterior illumination for grading cortical and/or PSC cataracts. The mean duration of the measurement by the SEC was 30.38 ± 6.27 s per 2 eyes, and the mean size of the filmed video was 85.96 ± 18.61 MB per 2 eyes (Table 1). Representative photos taken with the SEC are shown in Figure 3.

### 3.2. Correlation of Cataract Grading Evaluation by the Two Devices

The NUC value from the conventional slit-lamp microscope and the SEC showed a significantly strong correlation in the right eye (r = 0.926 [95%CI: 0.881 to 0.955; *p* < 0.001]; Table 2), left eye (r = 0.836 [95%CI: 0.743 to 0.898; *p* < 0.001]; Table 2), and both eyes (r = 0.871 [95%CI: 0.821 to 0.907; *p* < 0.001]; Table 2).

### 3.3. Reproducibility of the Cataract Grading Evaluated by the Two Devices

A high kappa value was observed between the cataract grading by the conventional slit-lamp microscope and that by the SEC (Kappa = 0.807 [95%CI: 0.798 to 0.816; *p* < 0.001]; Table 3).

## 4. Discussion

In the current study, we evaluated the cataract diagnosis by the new portable and recordable slit-lamp device, the “SEC.” The diagnostic accuracy and the performance of this device were compared with those of the conventional slit-lamp microscope. When we compared the NUC grading between the conventional slit-lamp microscope and the SEC, we found no significant differences of the gradings between the two devices (Table 1). Moreover, there was a significantly strong correlation between the NUC grading of the conventional slit-lamp and SEC evaluations (Table 2). Furthermore, using weighted kappa statistics to assess cataract grading demonstrated a high kappa value [11], which suggested that the images obtained by both devices were highly reproducible (Table 3). These results demonstrated that the performance of the SEC is equivalent to that of the conventional slit-lamp microscope in evaluating cataract. On the other hand, we did not estimate statistical power due to lack of past references. To address this disadvantage, we designed our study to screen every patient who visited a single hospital during a certain period prospectively. Our functional analysis suggested that the slit-light converted by the SEC allows evaluation of the severity of the NUC grading in cataract eyes, equal to that done using the conventional slit-lamp microscope.

Currently, few devices that can record anterior segment images by slit-light are available, although similar technologies have previously been reported. Some studies have reported the usefulness of slit-lamp microscope accessories that can attach to a smartphone camera. Chen et al. demonstrated good reproducibility of cataract grading using the same diagnosis criteria as those used in our study [12]. Moreover, Dubbs et al. reported a good rust ring image on the cornea taken using a smartphone [13]. These devices are attachable to the conventional non-portable slit-lamp microscopes, but not to smartphones. Mohammadpour et al. demonstrated the effectiveness of images obtained with a smartphone combined with a macro lens [14], using a concept similar to that of the SEC. However, the light source of the smartphone could release only diffuse white light, which was only capable of illuminating the surface of the eye. Chiong et al. invented a smartphone-based anterior segment-examining device that could irradiate the eyes with thick slit-light [15]. However, in ophthalmology departments, specialists require narrow slit-light that is thinner than the pupil width of 2‒8 mm [16] to ensure that the slit-light reaches the crystalline lens. To the best of our knowledge, no previous paper has reported that cataract grading was possible with a smartphone slit-lamp device. This may be due to the difficulty in converting the smartphone light source to a thin slit-light. The SEC has a function to overcome this problem by applying the slit structure above the light source and the cylinder lens to concentrate the light on the object. Therefore, we believe that SEC has an advantage over the conventional devices.

In this study, we demonstrated the usefulness of the SEC in cataract grading. The SEC may have the following potential advantages in healthcare: (1) It could be applied to other ophthalmological diseases, particularly in the anterior segment of the eye. We previously reported that this device could evaluate tear film breakup time and corneal epithelial disorder in a dry eye disease mouse model [8]. The size of the mouse eye is about 3-mm wide [17] and is much smaller than that in humans. Therefore, the SEC has the potential to be useful for the screening of other anterior segment eye diseases. (2) It could be used outside of the eye clinic. In the current study, an orthoptist rather than an ophthalmologist filmed the patient’s eye using the SEC. The average recording time was approximately 30 s for 2 eyes, which illustrates that the SEC is user-friendly for healthcare workers. Although the non-specialists can use this portable slit-lamp device, the ophthalmologists may need to evaluate the recorded images to make a cataract diagnosis. SEC is a smartphone attachment that allows video recording [8], and thus, it may be useful in telemedicine. Several studies have demonstrated that ophthalmological diagnoses are amenable to remote medicine, as the diagnosis is made based on images of the eye [18,19]. To the best of our knowledge, there is no other portable and recording device that can take a distinct eye photo with a thin slit-light. Therefore, the SEC may allow screening for and remote diagnosis of a cataract outside of an eye clinic. The challenge for the future is to verify whether or not the same results can be obtained when the product is actually used in the different fields including overseas.

In the current study, we used the WHO Cataract Grading Group for evaluating NUC gradings. Although the Lens Opacities Classification system III (LOCS III) is the most commonly used criteria, it mainly involves evaluation based on a specialist’s subjective assessment. Moreover, the subjective grading may change according to the specialists’ environment [20]. Thus, we did not use LOCS III.

However, it is true that not only NUC cases but also mixed type cases are common in actual clinical practice. Thus, the limitation of this study was that it included only NUC and not COR, ASC, and PSC. Since other cataract variations could also associate with reductions of visual acuity, we will evaluate in mixed type cases in the future study. However, NUC is the most common type especially in the elderly ages [21]. Therefore, we selected NUC for our primary evaluation. Moreover, our study included cases over 53 years of age and NUC grade 1 to 3 which were similar population to previous reports [22]. Our findings justify conduct of similar trials on a larger number of subjects with other types of cataract which will definitely provide invaluable information.

## 5. Conclusions

The results of this study suggested that not only the conventional, non-portable slit-lamp microscope, but also the portable and recordable slit-lamp, can diagnose and record cataract images appropriately. 

## 6. Patents

OUI, Inc. has the patent for the Smart Eye Camera (Japanese Patent No. 6627071. Inventors: H.Y., E.S., and N.A., Tokyo, Japan). There are no other relevant declarations relating to this patent.

## Figures and Tables

**Figure 1 diagnostics-10-00576-f001:**
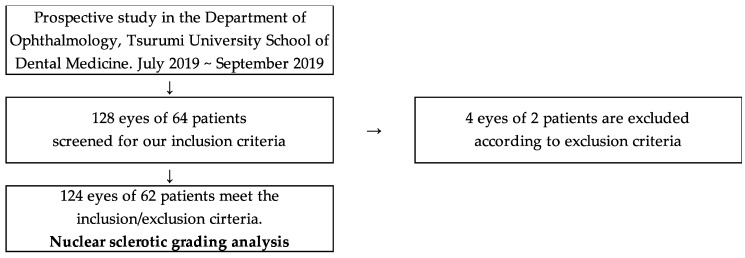
Study flowchart.

**Figure 2 diagnostics-10-00576-f002:**
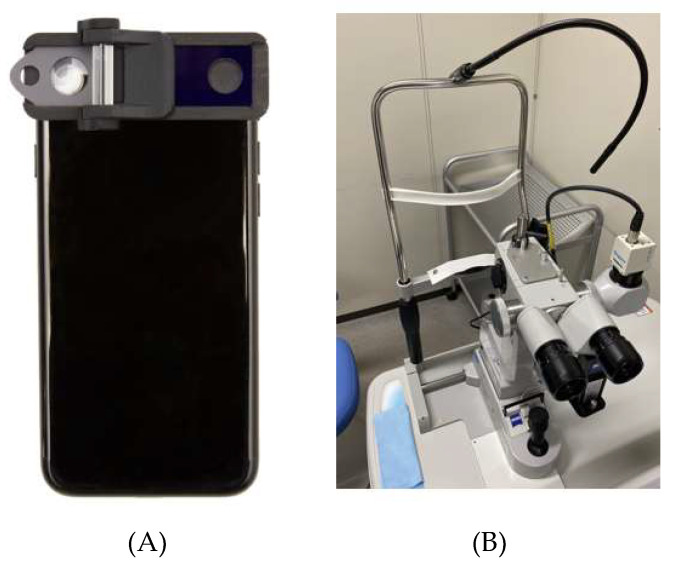
Appearance of (**A**) the SEC and (**B**) the conventional slit-lamp microscope.

**Figure 3 diagnostics-10-00576-f003:**
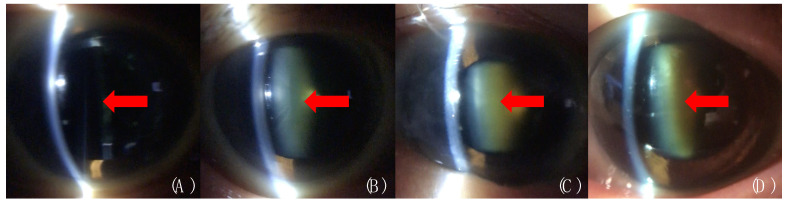
Representative cases imaged with the Smart Eye Camera. Eyes with (**A**) the lens replaced by an intraocular lens (IOL), (**B**) NUC grade 1, (**C**) NUC grade 2, and (**D**) NUC grade 3 are shown. Each red arrow indicates IOL or NUC.

**Table 1 diagnostics-10-00576-t001:** Demographic of the subjects.

Cases	64	
Male/Female	30/34	
Age	73.95 ± 9.28	
Eyes	128	
Phakia	110	
Pseudophakia	14	
Lack of data	4	
Nuclear Sclerotic grading		
Conventional/SEC	1.84 ± 0.82/1.92 ± 0.82	0.47 *
SEC		
Examination time, seconds	30.38 ± 6.27	
File size, MB	85.96 ± 18.61	

Data shown as mean ± SD. SEC: Smart Eye Camera. MB: Megabyte. * *p* value, Mann-Whitney’s U test.

**Table 2 diagnostics-10-00576-t002:** Correlation of the cataract grading evaluated by the two devices.

Eye	n	R	*p* Value *	95% CI	
R	62	0.926	< 0.001	0.881	0.955
L	62	0.836	< 0.001	0.743	0.898
Total	124	0.871	< 0.001	0.821	0.907

* Spearman’s correlation coefficient.

**Table 3 diagnostics-10-00576-t003:** Reproducibility of the cataract grading evaluated by the two devices.

		Smart Eye Camera		
**Conventional microscope**	Grade	1	2	3
	1	4	0	0
	2	5	78	2
	3	0	5	16
*p* Value		< 0.001		
Weighted kappa 95%CI		0.807		
		0.798–0.816		

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
