# Peer review of "Evaluation of Nuclear Cataract with Smartphone-Attachable Slit-Lamp Device"

_diagnostics, 2020, doi:10.3390/diagnostics10080576_

Round 1

Reviewer 1 Report

This study is an interesting report that developed a simple and inexpensive method of taking an anterior segment photograph using a smartphone to evaluate nuclear cataract.

   However, it seems that this method can be used only for grading the nuclear cataracts, and it is doubtful whether it can be widely used for diagnosis and cataract classification. Most cataracts are of the mixed type, and there are few cases of nuclear cataract alone, which makes them less versatile. If this device can be analyzed the other type of cataract, it will be beneficial.

Author Response

We hereby submit our revised manuscript to Diagnostics. We thank the editors and the reviewer for the interest in our manuscript and the constructive comments. We have given careful consideration to following comment and are grateful for the opportunity to improve our manuscript.

Response

Thank you for your excellent comment and agree with you. As you pointed out, in actual clinical practice, we generally diagnose cataracts not only NUC but also other types of cataracts. ASC and PSC can cause substantial visual impairment when they grow in the axial region of the lens. In particular, PSC is thought to cause significant visual impairment, particular in younger individuals and is most frequently the reason for surgery. However, the cause of cataract surgery in the elderly population is most commonly NUC, which was the only type of cataract evaluated in this study. Moreover, NUC is the most common type of cataract and it leads not only to visual impairment but also to poor depth perception and low-contrast sensitivity. Additionally, because numbers of ASC and PSC were very low in this study, we should collect more cataract cases and assess them in the future. Therefore, we selected NUC for our primary evaluating study and named this article “nuclear cataracat”. So, please refer to following sentence;

Page7, Line230-235 ; Our study only included NUC and did not include COR, ASC, and PSC. Other cataract variations could associate with reductions of visual acuity. However, NUC is the most common type especially in the elderly ages [21]. Therefore, we selected NUC for our primary evaluation. Moreover, our study included cases over 53 years of age and NUC grade 1 to 3 which were similar population to previous report [22]. Our findings justify conduct of similar trials on a larger number of subjects with other types of cataract which will definitely provide invaluable information.

Reviewer 2 Report

Dear authors,

Diagnostics Journal

Manuscript Number: diagnostics-878683

Title: Evaluation of Nuclear Cataract with Smartphone-Attachable Slit-Lamp Device

GENERAL CONSIDERATIONS

  • The authors have compared the performance of a portable Smartphone-Attachable Slit-Lamp Device to diagnose and classify cataract when compared to the conventional slit lamp.
  • Data is very interesting but some structural key points must be reviewed.

ABSTRACT

  • The authors have mentioned as purpose of the ABSTRACT: “Visual impairments and age-related eye diseases need to be detected and treated in a timely manner; however, this is often hampered by lack of appropriate medical equipment. We have invented a portable, recordable, and smartphone-attachable slit-lamp device, called the Smart Eye Camera (SEC).
  • The authors have mentioned as conclusion of the ABSTRACT: “This study suggests that the SEC is as reliable as the conventional non-portable slit-lamp microscope for evaluating NUC.
  • The purpose of the study is not cited in the BACKGROUND. Please add the purpose of the study in the background also.

METHODS

  • The authors have mentioned: It is a Japanese pharmaceuticals and medical devices agency-certified medical device that is able to convert the light source of the smartphone to a thin slit-light, allowing the light to enter the non-dilated pupil and reach the crystalline lens (Japanese Medical Device approval number: 13B2X10198030101).Confusing information. Please re-write the whole paragraph.
  • The authors have mentioned: It Although the Lens Opacities Classification system III is the most commonly used criteria, it mainly involves evaluation based on a specialist’s subjective assessment. Moreover, the subjective grading may change according to the specialists’ environment [11].” This sentence seem to be justifying the method used (why not LOCS classification?). The method should only contain the structured reproducible method used by the authors to obtain the results. Any justification should be mentioned in the DISCUSSION.

RESULTS

  • The authors have mentioned: In the current study, 128 eyes of 64 individuals (30 males and 34 females) were enrolled. Four eyes were excluded because the complete data was missing. The mean age of participants was 73.95 ± 9.28 years. In the current study, 110 of 124 eyes were phakic, 14 of 124 eyes were pseudophakic, and no aphakic eye was observed.” Please understand that this is MATERIAL studied. It will probably be more appropriate to mention this paragraph in its proper place (MATERIAL AND METHOD) unless this is part of the purpose of the study (which does not make any sense)
  • I understand that this article and journal have a general public as a target reader. If so, Figure 3 must contain more specific explanation (maybe with arrows and colors). Please choose to add this explanation to the general public in the result (if the authors think it is part of the purpose) or in the material (if the authors think it is part of the method)
  • The authors have cited in line 156: To compare the performance of cataract grading by the conventional slit-lamp microscope and the SEC, the correlation of the NUC grading was evaluated.” Please read this carefully. This sentence seems to be explaining how data was collected and evaluated. So this is part of the method. Please provide it in its proper place (MATERIAL AMD METHOD).
  • The authors have cited in line 163: To assess the reproducibility of the cataract grading by the conventional slit-lamp microscope and by the SEC, weighted kappa statistics of the NUC grading were evaluated.” Please read this carefully. This sentence seems to be explaining how data was collected and evaluated. So this is part of the method. Please provide it in its proper place (MATERIAL AMD METHOD).

DISCUSSION

  • The authors have cited as purpose (line 56): Here, we verified the performance of SEC by the clinical trial, in which we evaluated and compared the efficacy and diagnostic performance for cataract diagnosis with those of a conventional slit-lamp microscope.”
  • The authors have cited as conclusion: The results of this study suggested that not only the conventional, non-portable slit-lamp microscope, but also the portable and recordable slit-lamp can diagnose and record cataract images appropriately. An ophthalmologist successfully evaluated the videos taken by a non-ophthalmologist using the device. It may hold great potential for the diagnosis of other ophthalmological diseases in the anterior segment of the eyes, application outside of the eye clinic, and application in telemedicine. The challenge for the future is to verify whether or not the same results can be obtained when the product is actually used in the different fields including overseas.
  • Adding the word “clinical trial” (in line 56) might confuse the readers and maybe require clinical trial registration. I would rather delete this word. It will not make any different on the sentence.
  • The authors have mentioned: “An ophthalmologist successfully evaluated the videos taken by a non-ophthalmologist using the device.” This is a very interesting point and a very smart issue. However, it is not part of the purpose of the article. Due the importance of the fact, I would add it as a secondary purpose of the article and show some data of non-ophthalmologist capturing the image.
  • The authors have mentioned: “It may hold great potential for the diagnosis of other ophthalmological diseases in the anterior segment of the eyes, application outside of the eye clinic, and application in telemedicine. The challenge for the future is to verify whether or not the same results can be obtained when the product is actually used in the different fields including overseas.” This is not part of the purpose of the article; I would rather delete this sentence and cite this in another paragraph in the DISCUSSION.

Author Response

We hereby submit our revised manuscript to Diagnostics. We thank the editors and the reviewer for the interest in our manuscript and the constructive comments. We have given careful consideration to all comments and are grateful for the opportunity to improve our manuscript.

Comment 1

The authors have mentioned as purpose of the ABSTRACT: “Visual impairments and age-related eye diseases need to be detected and treated in a timely manner; however, this is often hampered by lack of appropriate medical equipment. We have invented a portable, recordable, and smartphone-attachable slit-lamp device, called the Smart Eye Camera (SEC).”

The authors have mentioned as conclusion of the ABSTRACT: “This study suggests that the SEC is as reliable as the conventional non-portable slit-lamp microscope for evaluating NUC.”

The purpose of the study is not cited in the BACKGROUND. Please add the purpose of the study in the background also.

Response 1

Thank you for your excellent comment and agree with you. We added the sentence of the purpose of this study as follows;

Page1, Line19-22 ;・・・The aim of this study was to compare evaluating nuclear cataract (NUC) between the SEC and the conventional, non-portable slit-lamp microscope. Methods: A total of 128 eyes of 64 Japanese patients (mean age: 73.95 ± 9.28 years; range: 51‒92 years; female: 34) were enrolled. ・・・

Comment 2

The authors have mentioned: “It is a Japanese pharmaceuticals and medical devices agency-certified medical device that is able to convert the light source of the smartphone to a thin slit-light, allowing the light to enter the non-dilated pupil and reach the crystalline lens (Japanese Medical Device approval number: 13B2X10198030101).” Confusing information. Please re-write the whole paragraph.

Response 2

Thank you for your great comment and sorry for confusing you. We revised the sentence as follows;

Page3, Line92-94 ;・・・The device has been approved as a medical device in Japan (Japanese Medical Device approval number: 13B2X10198030101), and this is able to convert the light source of the smartphone to a thin slit-light, allowing the light to reach the crystalline lens.・・・

Comment 3

The authors have mentioned: “It Although the Lens Opacities Classification system III is the most commonly used criteria, it mainly involves evaluation based on a specialist’ssubjective assessment. Moreover, the subjective grading may change according to the specialists’ environment [11].” This sentence seem to be justifying the method used (why not LOCS classification?). The method should only contain the structured reproducible method used by the authors to obtain the results. Any justification should be mentioned in the DISCUSSION.

Response 3

Thank you for your great comment and agree with you. We moved the sentence to DISCUSSION section and revised as follows;

Page7, Line225-229 ;In the current study, we used the WHO Cataract Grading Group for evaluating NUC gradings. Although the Lens Opacities Classification system III (LOCS III) is the most commonly used criteria, it mainly involves evaluation based on a specialist’s subjective assessment. Moreover, the subjective grading may change according to the specialists’ environment [20]. Thus, we did not use LOCS III.

Comment 4

The authors have mentioned: “In the current study, 128 eyes of 64 individuals (30 males and 34 females) were enrolled. Four eyes were excluded because the complete data was missing. The mean age of participants was 73.95 ± 9.28 years. In the current study, 110 of 124 eyes were phakic, 14 of 124 eyes were pseudophakic, and no aphakic eye was observed.” Please understand that this is MATERIAL studied. It will probably be more appropriate to mention this paragraph in its proper place (MATERIAL AND METHOD) unless this is part of the purpose of the study (which does not make any sense).

Response 4

Thank you for your great comment and agree with you. We moved the sentence to MATERIALS AND METHODS section and revised as follows;

Page2, Line61-65; In this prospective study, 128 eyes of 64 individuals (30 males and 34 females) were enrolled. 4 eyes were excluded because the complete data was missing. The mean age of participants was 73.95 ± 9.28 years. In the current study, 110 of 124 eyes were phakic, 14 of 124 eyes were pseudophakic, and no aphakic eye was observed (Figure 1). This prospective study was performed according to the steps shown in Figure 1. We screened・・・

Comment 5

I understand that this article and journal have a general public as a target reader. If so, Figure 3 must contain more specific explanation (maybe with arrows and colors). Please choose to add this explanation to the general public in the result (if the authors think it is part of the purpose) or in the material (if the authors think it is part of the method)

Response 5

Thank you for your excellent suggestion and agree with you. We added arrows indicating NUC and its explanation. Please refer to Figure 3 and its revised sentences in new manuscript.

Page5, Line158-159; Each red arrow indicates an IOL or NUC.

Comment 6

The authors have cited in line 156: “To compare the performance of cataract grading by the conventional slit-lamp microscope and the SEC, the correlation of the NUC grading was evaluated.” Please read this carefully. This sentence seems to be explaining how data was collected and evaluated. So this is part of the method. Please provide it in its proper place (MATERIAL AND METHOD).

Response 6

Thank you for your great comment and agree with you. We deleted the sentence from RESULTS section, and added into MATERIALS AND METHODS section as follows;

Page4, Line136-139; ・・・To assess the reproducibility of the cataract grading by the two devices (conventional slit-lamp microscope and SEC), weighted kappa statistics were selected. Moreover, Spearman's correlation coefficient was used to assess the correlation of the cataract grading evaluated using the 2 devices. Data were・・・

Comment 7

The authors have cited in line 163: “To assess the reproducibility of the cataract grading by the conventional slit-lamp microscope and by the SEC, weighted kappa statistics of the NUC grading were evaluated.” Please read this carefully. This sentence seems to be explaining how data was collected and evaluated. So this is part of the method. Please provide it in its proper place (MATERIAL AND METHOD).

Response 7

Thank you for your great comment and agree with you. We deleted the sentence from RESULTS section, and added into MATERIALS AND METHODS section as in Response 6. Please refer to Response 6.

Comment 8

The authors have cited as purpose (line 56): “Here, we verified the performance of SEC by the clinical trial, in which we evaluated and compared the efficacy and diagnostic performance for cataract diagnosis with those of a conventional slit-lamp microscope.” The authors have cited as conclusion: “The results of this study suggested that not only the conventional, non-portable slit-lamp microscope, but also the portable and recordable slit-lamp can diagnose and record cataract images appropriately. An ophthalmologist successfully evaluated the videos taken by a non-ophthalmologist using the device. It may hold great potential for the diagnosis of other ophthalmological diseases in the anterior segment of the eyes, application outside of the eye clinic, and application in telemedicine. The challenge for the future is to verify whether or not the same results can be obtained when the product is actually used in the different fields including overseas.” Adding the word “clinical trial” (in line 56) might confuse the readers and maybe require clinical trial registration. I would rather delete this word. It will not make any different on the sentence.

Response 8

Thank you for your great comment and sorry for confusing you. We agree with your pointing out, so we changed the sentence to be more concise and easier to understand as follows;

Page2, Line56-58 ;・・・Here, we evaluated and compared the effectiveness and diagnostic performance for cataract diagnosis between SEC and conventional slit lamp microscope.

Comment 9

The authors have mentioned: “An ophthalmologist successfully evaluated the videos taken by a non-ophthalmologist using the device.” This is a very interesting point and a very smart issue. However, it is not part of the purpose of the article. Due the importance of the fact, I would add it as a secondary purpose of the article and show some data of non-ophthalmologist capturing the image.

Response 9

Thank you for your great comment and agree with you. But it was very invaluable for us. We mentioned it in the DISCUSSION section, so we removed the text from the CONCLUSIONS section so as not to duplicate it.

Page7, Line213-219 ;・・・In the current study, an orthoptist rather than an ophthalmologist filmed the patient's eye using the SEC. The average recording time was approximately 30 seconds for 2 eyes, which illustrates that the SEC is user-friendly for healthcare workers. Although the non-specialists can use this portable slit-lamp device, the ophthalmologists may need to evaluate the recorded images to make a cataract diagnosis. SEC is a smartphone attachment that allows video recording [8], and thus, it may be useful in telemedicine. Several・・・

Comment 10

The authors have mentioned: “It may hold great potential for the diagnosis of other ophthalmological diseases in the anterior segment of the eyes, application outside of the eye clinic, and application in telemedicine. The challenge for the future is to verify whether or not the same results can be obtained when the product is actually used in the different fields including overseas.” This is not part of the purpose of the article; I would rather delete this sentence and cite this in another paragraph in the DISCUSSION.

Response 10

Thank you for your great comment and agree with you. We mentioned it in the DISCUSSION section and revised the sentences in CONCLUSIONS section as follows;

 ãƒ»Page7, Line223-224 ;・・・eye clinic. The challenge for the future is to verify whether or not the same results can be obtained when the product is actually used in the different fields including overseas.

・Page7, Line237-243 ; The results of this study suggested that not only the conventional, non-portable slit-lamp microscope, but also the portable and recordable slit-lamp can diagnose and record cataract images appropriately. An ophthalmologist successfully evaluated the videos taken by a non-ophthalmologist using the device. It may hold great potential for the diagnosis of other ophthalmological diseases in the anterior segment of the eyes, application outside of the eye clinic, and application in telemedicine. The challenge for the future is to verify whether or not the same results can be obtained when the product is actually used in the different fields including overseas.

Round 2

Reviewer 1 Report

As mentioned in the first review, there are not many cataracts with nuclear opacity alone, and this simple imaging method using smartphone that can detect and analyze only nuclear cataracts is considered to be not significant effective.

Author Response

We hereby submit our revised manuscript to Diagnostics. We thank the reviewer 1 for the interest in our manuscript and the constructive comments. We have given careful consideration to following comment and are grateful for the opportunity to improve our manuscript.

Comment 1

As mentioned in the first review, there are not many cataracts with nuclear opacity alone, and this simple imaging method using smartphone that can detect and analyze only nuclear cataracts is considered to be not significant effective.

Response 1

Thank you for your excellent comment and agree with you. The academic editor said the same thing. Therefore, we added following sentence;

Page7, Line230-238 ; However, it is true that not only NUC cases but also mixed type cases are common in actual clinical practice. Thus, the limitation of this study was that it included only NUC and not COR, ASC, and PSC. Because other cataract variations could also associate with reductions of visual acuity, we will evaluate in mixed type cases in the future study. However, NUC is the most common type especially in the elderly ages [21]. Therefore, we selected NUC for our primary evaluation. Moreover, our study included cases over 53 years of age and NUC grade 1 to 3 which were similar population to previous report [22]. Our findings justify conduct of similar trials on a larger number of subjects with other types of cataract which will definitely provide invaluable information.